# Dynamic density shaping of photokinetic *E. coli*

Giacomo Frangipane[1], Dario Dell'Arciprete[1], Serena Petracchini[2], Claudio Maggi[3], Filippo Saglimbeni[3], Silvio Bianchi[3], Gaszton Vizsnyiczai[1], Maria Lina Bernardini[2,4], Roberto Di Leonardo[1,3]*

[1]Dipartimento di Fisica, Università di Roma "Sapienza", Roma, Italy; [2]Istituto Pasteur-Fondazione Cenci Bolognetti, Università di Roma "Sapienza", Roma, Italy; [3]Soft and Living Matter Laboratory, Institute of Nanotechnology (NANOTEC-CNR), Roma, Italy; [4]Dipartimento di Biologia e Biotecnologie, Università di Roma "Sapienza", Roma, Italy

**Abstract** Many motile microorganisms react to environmental light cues with a variety of motility responses guiding cells towards better conditions for survival and growth. The use of spatial light modulators could help to elucidate the mechanisms of photo-movements while, at the same time, providing an efficient strategy to achieve spatial and temporal control of cell concentration. Here we demonstrate that millions of bacteria, genetically modified to swim smoothly with a light controllable speed, can be arranged into complex and reconfigurable density patterns using a digital light projector. We show that a homogeneous sea of freely swimming bacteria can be made to morph between complex shapes. We model non-local effects arising from memory in light response and show how these can be mitigated by a feedback control strategy resulting in the detailed reproduction of grayscale density images.
DOI: https://doi.org/10.7554/eLife.36608.001

*For correspondence:
roberto.dileonardo@uniroma1.it

Competing interests: The authors declare that no competing interests exist.

## Introduction

A local reduction of speed in a crowd of walking pedestrians or urban car traffic usually results in the formation of high-density regions. Mathematically speaking, a self-propelled particle, moving by an isotropic random walk with space dependent speed, explores the available space with a probability density that is inversely proportional to the local value of the speed (*Schnitzer, 1993*; *Tailleur and Cates, 2009*; *Cates and Tailleur, 2015*), spending a longer time in slow regions than in fast ones. As a consequence, any mechanism that allows to substantially modulate the local propulsion speed can be used as an efficient strategy for controlling the density (*Stenhammar et al., 2016*). In swimming bacteria, proteorhodopsin (PR) provides a particularly convenient way of achieving spatial speed control through the projection of light patterns. PR is a light-driven proton pump that uses photon energy to pump protons out of the cytoplasm, thus modulating the corresponding electrochemical gradient across the inner membrane (*Béjà et al., 2000*). Since this proton motive force drives the rotation of the flagellar motor (*Gabel and Berg, 2003*), PR puts a 'solar panel' on every cell allowing to remotely control its swimming speed with light (*Walter et al., 2007*). This has been recently exploited to control the rotational speeds of bio-hybrid micromachines using bacteria as micropropellers (*Vizsnyiczai et al., 2017*). In light driven Janus colloids, spatially asymmetric speed profiles can also affect particle orientation giving rise to an artificial phototaxis mechanism (*Lozano et al., 2016*). *Arlt et al. (2018)* have recently shown that, by projecting a masked illumination pattern, 'photokinetic' (*Wilde and Mullineaux, 2017*) bacteria can be accumulated in dark regions and depleted from brighter ones thus forming binary patterns. Here we demonstrate that bacterial density can be controlled to form reconfigurable complex patterns. To do this we exploit a Digital Light

**eLife digest** Many bacteria can move in response to environmental signals. This helps guide them towards better conditions for growth and survival. *Escherichia coli* is a bacterium that can swim quickly through liquid, using tiny propeller-like structures that rotate many times per second. These 'propellers' are powered by a cellular motor, called the flagellar motor, which similar to an electric motor, requires an energy source to drive movement.

Proteorhodopsin, a protein originally isolated from free-swimming micro-organisms in the ocean, is an alternative energy source that helps bacteria move. The protein is located close to the surface of the cell, where it acts like a solar panel and captures energy from light. In cells powered by proteorhodopsin, the intensity of light from their environment determines their swimming speed: brighter light means faster movement, and less light, slower movement. Proteorhodopsin is now also a useful tool in the laboratory. For example, genetically engineering bacteria to produce proteorhodopsin provides a way to control their movement remotely, using a light source.

Swimming bacteria, much like cars in city traffic, are known to accumulate in areas where their speed decreases. By controlling swimming speed with proteorhodopsin, researchers can manipulate the local density of bacteria simply by projecting different patterns of light.

To study the factors influencing this phenomenon, Frangipane et al. used genetically modified *E. coli* that could respond to light via proteorhodopsin to make layers of cells that could then have light patterns projected onto them. The results showed that the bacteria responded slowly to these stimuli, which was the main factor limiting the resolution of the final pattern they formed. A simple feedback mechanism, which compared the pattern formed by the cells to the desired image and updated the projected light accordingly, was enough to solve this problem. This way, the layers of *E.coli* could be turned into a near-perfect copy of the original image.

This work allows us to control the movement of large populations of bacteria more precisely than ever before. This could be extremely valuable for building the next generation of microscopic devices. For example, bacteria could be made to surround a larger object such as a machine part or a drug carrier, and then used as living propellers to transport it where it is needed.
DOI: https://doi.org/10.7554/eLife.36608.002

Processing (DLP) projector to shape light with a megapixel resolution and with a dynamic range of 256 (8 bit) light intensity levels. We experimentally investigate the limits of validity of the predicted law of inverse proportionality between local density and speed. We demonstrate that observed deviations from this law can be quantitatively described by a theoretical model that explicitly takes into account memory effects in light response. Finally we show that a model independent feedback control loop allows density shaping with high spatial resolution and gray level accuracy.

## Results

Density shaping in photokinetic bacteria or colloids relies on the connection between density and local speed. The density $\rho(\mathbf{r}) \propto 1/v(\mathbf{r})$ is always a steady-state solution of the master equation of a self-propelled particle with isotropic reorientation dynamics and spatially varying speed $v(\mathbf{r})$ (*Cates and Tailleur, 2015*). In our case, $v(\mathbf{r}) = v(I(\mathbf{r}))$ where $\mathbf{r} = (x, y)$ is the position in the two dimensional image space and $I(\mathbf{r})$ the light intensity at $\mathbf{r}$. This result assumes that all bacteria have the same response to light and that they instantaneously adapt to intensity variations so that the speed only depends on the local value of $I$. However, suspensions of swimming bacteria, are known to be characterized by motility properties that occur in broad distributions and, in principle, a broad spectrum of light responses could be found. To quantify the effect of light on the speed distributions of our strain, we monitored bacterial dynamics while projecting a chessboard pattern consisting of 12 different levels of light intensity. To do this we couple a DLP projector to a custom video-microscopy setup working in both bright-field and dark-field mode (see *Figure 1* and Materials and methods).

Using differential dynamic microscopy (DDM) (*Cerbino and Trappe, 2008*; *Wilson et al., 2011*; *Maggi et al., 2013*) we extract the light dependent values of the mean $v$ and standard deviation $\sigma$ of the speed distributions (*Figure 2(a)*) (see Materials and methods). Although the absolute value of

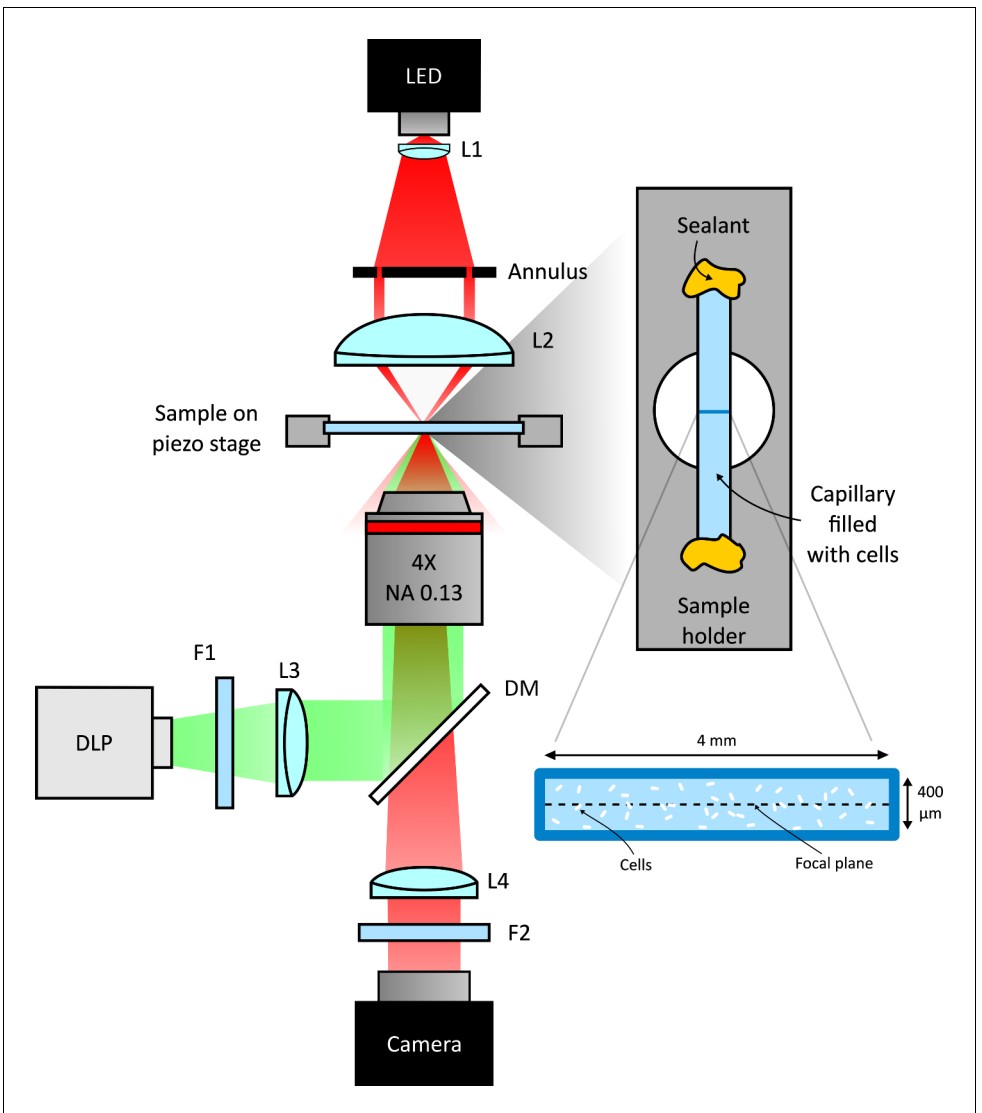

**Figure 1.** Scheme of the experimental setup. Schematic representation of the custom inverted-microscope used in the experiments (see Materials and methods). Green light from a DMD based DLP projector is filtered by a bandpass filter F1 (central wavelength 520 nm) and than coupled to the microscope objective through a dichroic mirror (DM). A long pass filter (F2) prevents illumination light to reach the camera. Bacteria are sealed in a square microcapillary glued on a metallic sample holder with a circular aperture.

DOI: https://doi.org/10.7554/eLife.36608.003

the swimming speed depends on several factors (strain, growth medium, etc.) we find a non linear speed response that is consistent with what previously reported for PR expressing bacteria. In particular the speed versus intensity curve is very well fitted by a hyperbola showing saturation for large light intensity values as already reported in (*Walter et al., 2007*; *Schwarz-Linek et al., 2016*; *Vizsnyiczai et al., 2017*; *Arlt et al., 2018*).

We also find that, throughout the intensity range, $v$ and $\sigma$ are directly proportional (see *Figure 2 (b)*) suggesting a homogeneous growth law for individual cell speeds $v_i(I) = v_i^s f(I)$, where $v_i^s$ is the saturation speed for the $i$-th cell and $f(I)$ is a dimensionless function saturating at one for large intensities. Interestingly, in this scenario, the mean speed $v(I)$ will also be proportional to $f(I)$ so that the probability density $\rho_i$ for the generic $i$-th cell will be proportional to the inverse of the local mean speed:

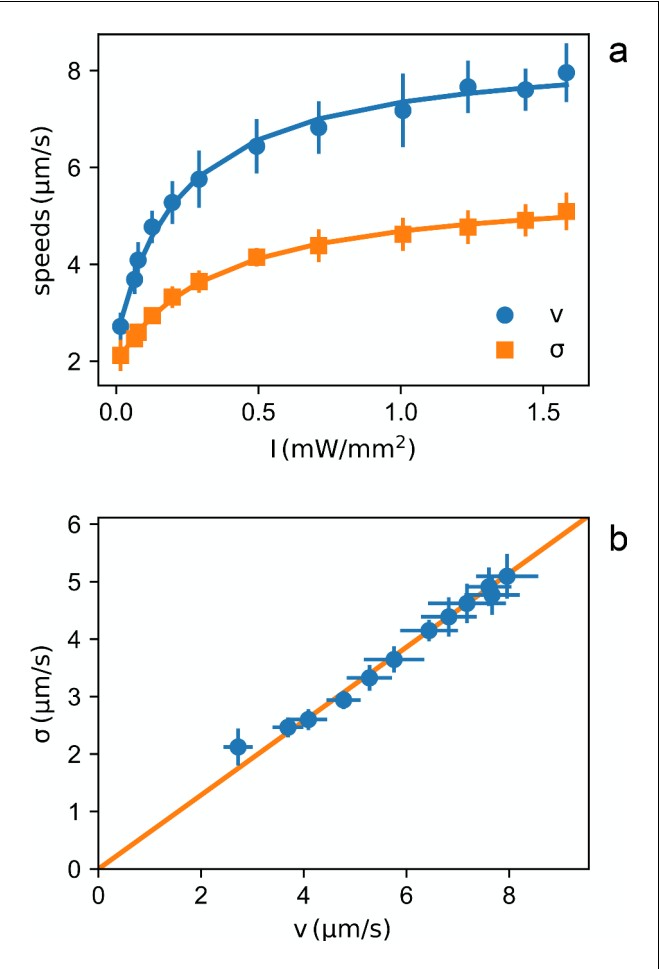

**Figure 2.** Light dependent speed distribution. (a) Mean value $v$ (circles) and standard deviation $\sigma$ (squares) of bacterial speeds as a function of light power density. Symbols and error bars are respectively the mean and standard deviation of 14 repeated measurements of $v$ and $\sigma$. Curves are fits with hyperbolas. (b) Parametric plot of $\sigma$ vs $v$. The line is a linear fit passing through zero.

DOI: https://doi.org/10.7554/eLife.36608.004

The following source data is available for figure 2:

**Source data 1.** Source data for *Figure 2*

DOI: https://doi.org/10.7554/eLife.36608.005

$$\rho_i(\mathbf{r}) \propto \frac{1}{v_i(I(\mathbf{r}))} \propto \frac{1}{f(I(\mathbf{r}))} \propto \frac{1}{v(I(\mathbf{r}))} \ .$$

This implies that, despite broad speed distributions, the local density is expected to be only determined by the mean local speed, as set by the light intensity pattern and measured by DDM. To validate this hypothesis we use a DLP projector to display a complex light pattern onto a 400 µm thick layer of cells (*Figure 1*) that have been preliminarily exposed to a uniform bright illumination for 5 min. This time is much longer than the speed response time of bacteria (Figure 4) thus ensuring that cells are initialized to swim at maximal speed. The projecting system has an optical resolution of 2 µm approximately matching the size of a single cell which represents the physical 'pixel' of our density images. This value sets the limit for the minimum theoretical resolution of density configurations that would be achievable if bacteria could be precisely and statically arranged in space. In practice, as we will see, the real resolution will always be larger for two main reasons: (i) bacteria do not respond instantly to light temporal variations thus introducing a blur in the target speed map, (ii) the

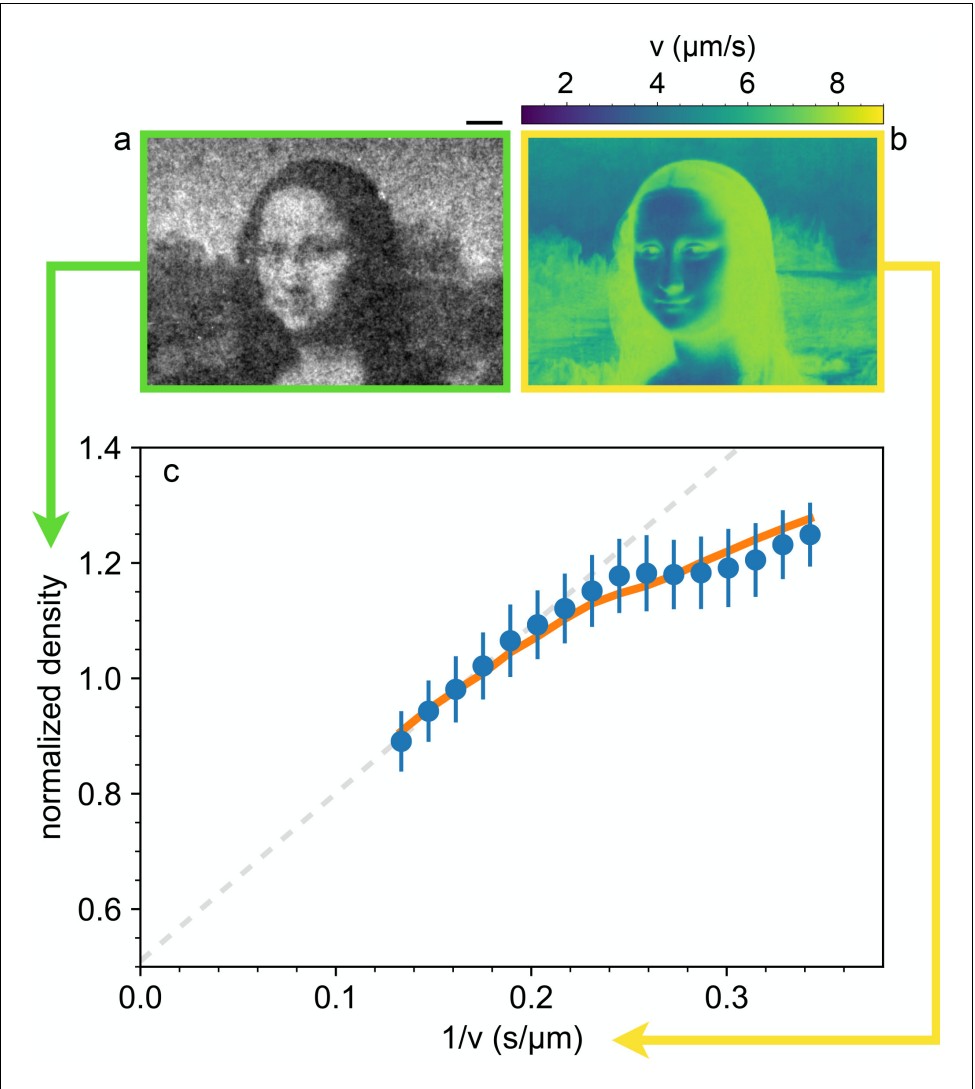

**Figure 3.** Shaping density with light. (**a**) Dark-field microscopy image of the sample obtained after projecting a static light pattern for 4 min (averaged for 2 min). Scale bar 100 μm. (**b**) Local speed map obtained by interpolating the data-points of *Figure 2(a)* at the values of the local (projected) light intensity pattern. (**c**) The circles represent the mean value of the normalized sample density over image pixels corresponding to the same light intensity, plotted as a function of the corresponding inverse local speed. The dashed line is a linear fit of the high-speed points. The error bars are the standard deviation of the density at the same value of the computed inverse average speed. Solid line represent the prediction from the memory blur model described in the text.
DOI: https://doi.org/10.7554/eLife.36608.006
The following source data is available for figure 3:

**Source data 1.** Source data for *Figure 3*
DOI: https://doi.org/10.7554/eLife.36608.007

stationary state is an ensemble of noisy patterns that constantly fluctuate because of swimming and Brownian motions of bacteria. After projecting an inverted Mona Lisa image, bacteria start to concentrate in dark regions while moving out from the more illuminated areas. After about 4 min, dark field microscopy reveals a recognizable bacterial 'replica' of Leonardo's painting, where brighter areas correspond to regions of accumulated cells. Once a stationary pattern is reached, we collect a time averaged image from which we extract the bacterial density $\rho^*$ shown in *Figure 3* normalized to be one when uniform (see Materials and methods).

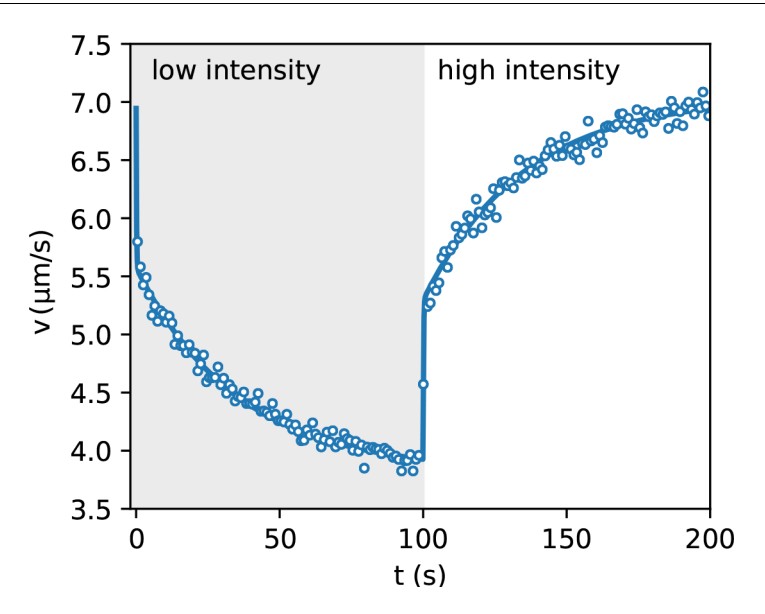

**Figure 4.** Speed response to light step. Average bacteria speeds as a function of time during periodic illumination with a square wave of period 200 s. Open circles represent experimental data averaged over eight periods while solid line is a fit with a step response followed by an exponential relaxation.

DOI: https://doi.org/10.7554/eLife.36608.008

The following source data is available for figure 4:

**Source data 1.** Source data for *Figure 4*

DOI: https://doi.org/10.7554/eLife.36608.009

We split the density image into components corresponding to the same illumination level $I$. We then average the density within each component and report in *Figure 3(c)* the obtained value as a function of the inverse mean speed $1/v(I)$ obtained from the speed *vs* intensity curve in *Figure 2(a)*. The high speed side of the graph can be very well fitted by a straight line with a finite intercept $q = 0.5$. This suggests that 50% of the total scattering objects in the field of view responds to light and can be spatially modulated while the remaining 50% can be attributed to cells which are non-motile or insensitive to light and also to stray light generated away from the focal plane. The ratio between the maximum and minimum modulation of the light sensitive component can be obtained as $max[\rho^* - q]/min[\rho^* - q] = 2$. A strong deviation from linearity is evident in the high density/low speed region. A violation of the $\rho \propto 1/v$ law can be attributed to many factors that are not included in the simple theory discussed above. In particular, the theory assumes that speed is a local function of space which would only be the case if bacteria instantly adapt to temporal changes in light intensity. However, previous studies have evidenced that speed response is not instantaneous but it displays a relaxation pattern characterized by multiple timescales. In addition to a fast relaxation time associated to the membrane discharge (*Walter et al., 2007*), ATP synthase and the dynamics of stator units of the flagellar motors introduce slower timescales in the speed relaxation dynamics (*Tipping et al., 2013*; *Arlt et al., 2018*). Using DDM we measured speed response to a uniform light pattern whose intensity switches between two values every 100 s. Results are reported in *Figure 4* and clearly show the presence of two timescales. The short time scale is smaller than our time resolution (1 s) and appears as an instantaneous jump that accounts for about a fraction $\beta = 0.44$ of the total relaxation. A slower relaxation follows and it is well fitted by an exponential function with a time constant $\tau_m = 35$ s that is the same for both the rising and falling relaxations. As a result, bacteria will experience an effective speed map that is a blurred version of what we would expect for an instantaneous response $V(\mathbf{r}) = v(I(\mathbf{r}))$. In the case of smooth swimming cells, with a two-step light response, we calculate that, for weak speed modulations around a baseline value $V_0$, the actual speed map can be obtained as a simple convolution (see Materials and methods):

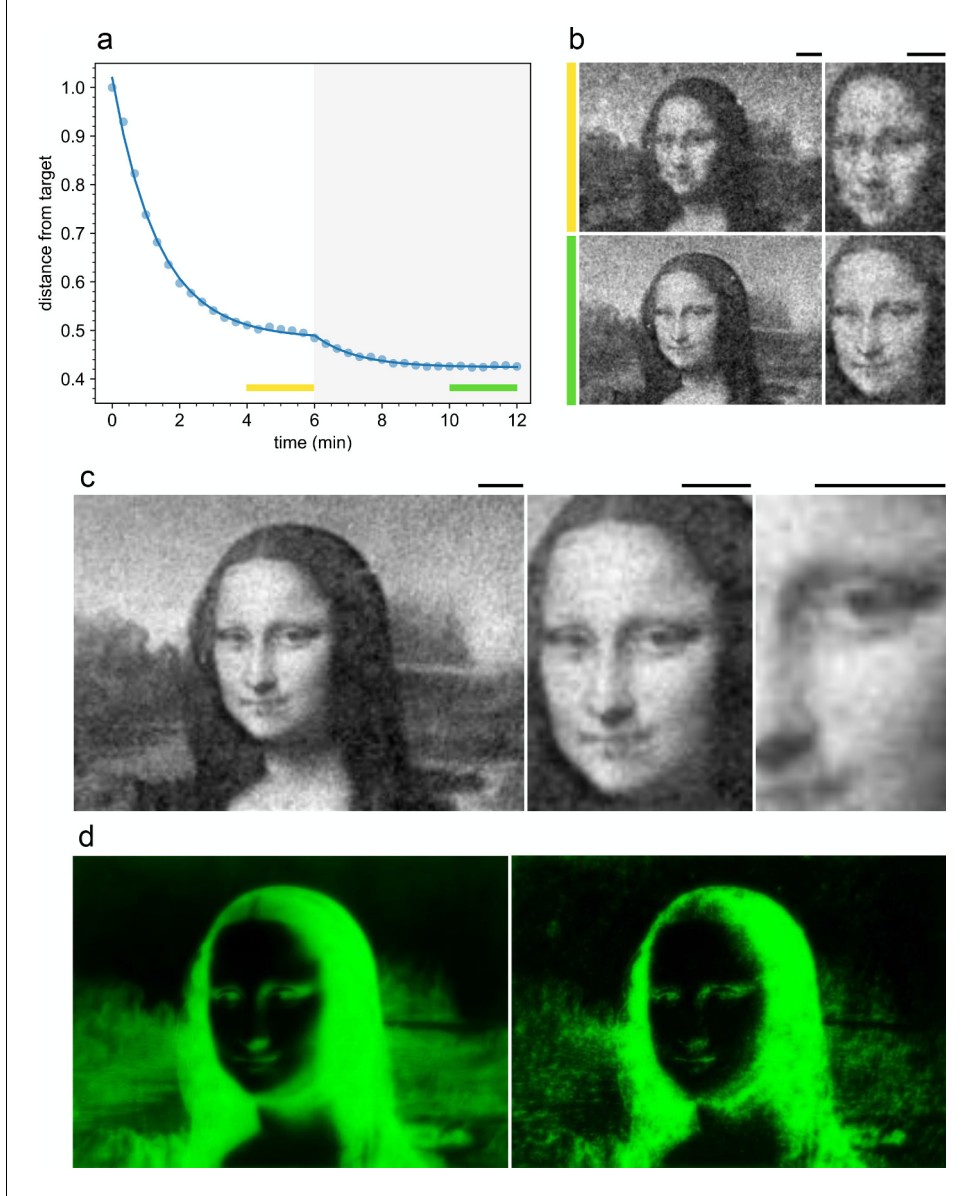

**Figure 5.** Improved density control with a feedback loop. (a) Time evolution of the distance from the target normalized to the initial value (circles) before and after activating the feedback loop (gray area). The yellow and green bars indicate the time interval over which we average the density maps (shown in (b)) before and after feedback respectively. The full curve is a fit with a double exponential. (b) Comparison of the density map obtained by averaging for 2 min before (top) and after the feedback loop has been turned on (bottom) (see colored bars in (a)). (c) Time averaged density profile (6 min) with feedback on. (d) Projected light intensity patterns at $t = 0$ (left) and after feedback optimization (right). Scale bars are 100 μm.

DOI: https://doi.org/10.7554/eLife.36608.010

The following source data is available for figure 5:

**Source data 1.** Source data for *Figure 5*

DOI: https://doi.org/10.7554/eLife.36608.011

$$w(x,y) \simeq \beta V(x,y) + (1-\beta) \int V(x-x', y-y')\gamma(x',y')dx'dy' \qquad (1)$$

with $\gamma(x,y)$ a convolution kernel that is analytic in Fourier space:

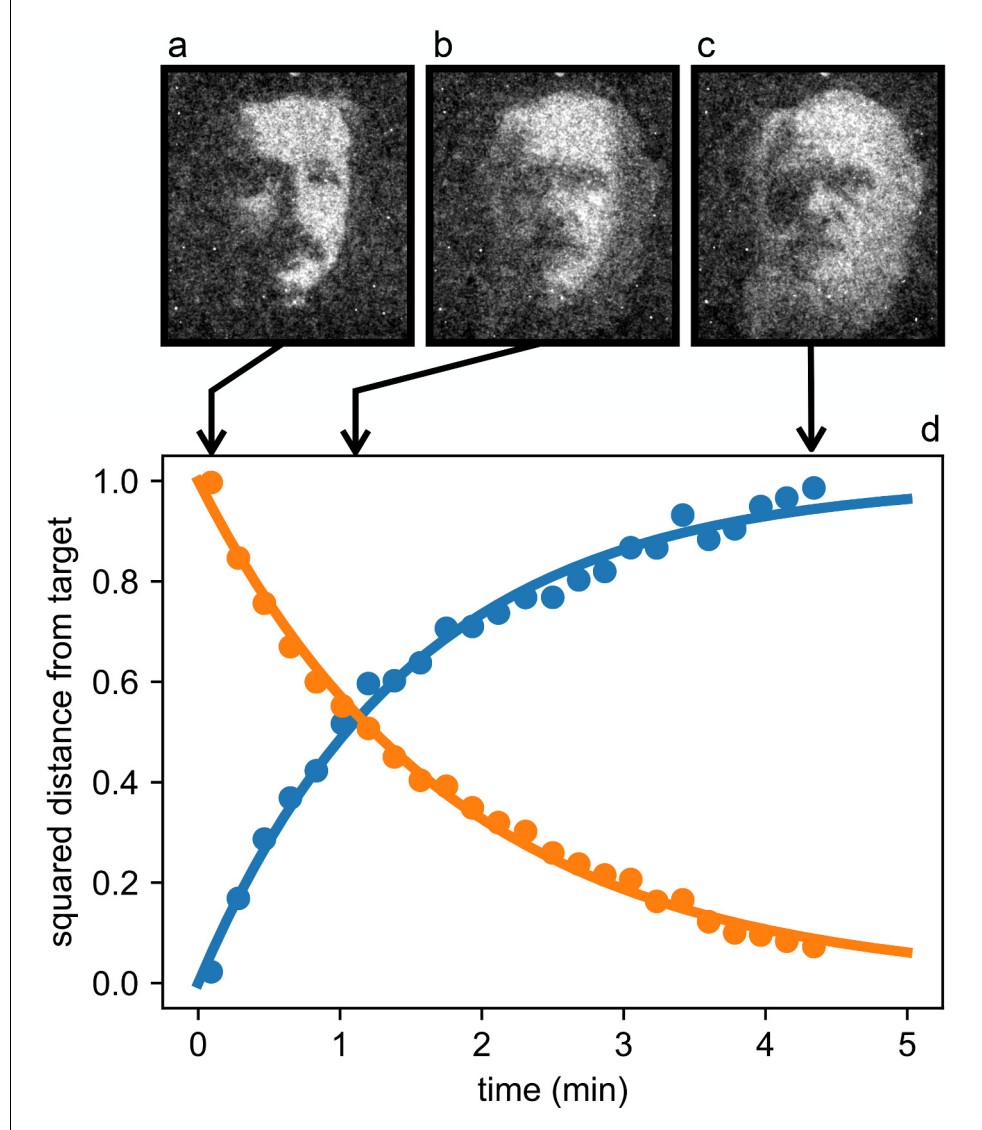

**Figure 6.** Reconfigurable density patterns. Starting from the stationary density modulation (**a**) we switch to a new light pattern at time 0 and record the density distribution of bacteria as it morphs through the intermediate state (**b**) and reaches the final state (**c**). (**d**) Time evolution of the normalized squared distances between instantaneous density maps and the initial (blue circles) and final (orange circles) targets. Curves are exponential fits.

DOI: https://doi.org/10.7554/eLife.36608.012

The following source data is available for figure 6:

**Source data 1.** Source data for *Figure 6*

DOI: https://doi.org/10.7554/eLife.36608.013

$$\widetilde{\gamma}(q) = \frac{k}{q}\arctan\left(\frac{q}{k}\right) \tag{2}$$

where $q = \sqrt{q_x^2 + q_y^2}$ is the modulus of the wave vector and $k^{-1} = v_0\tau_m$. Assuming that the stationary distribution will remain isotropic even in the presence of memory, the relation $\rho \propto 1/w$ will then still be valid provided one uses the actual, blurred speed map $w$ and not the original map $V$ corresponding to instantaneous response. We can then anticipate the stationary density once we calculate the

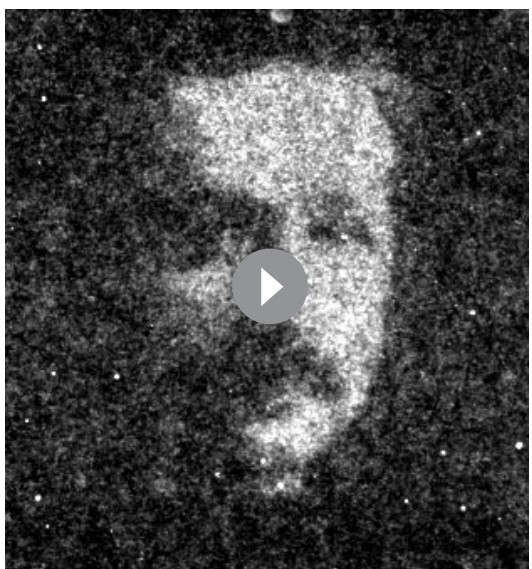

**Video 1.** The video shows the dynamic morphing of a layer of photokinetic bacteria from an Albert Einstein's to a Charles Darwin's portrait (total time 5 min).
DOI: https://doi.org/10.7554/eLife.36608.014

actual speed map $w$ with the convolution formula (1). Recalling that only a fraction $\alpha$ of cells is motile the estimated normalized stationary density will be

$$\rho^*(x,y) = \alpha\left(\frac{w(x,y)^{-1}}{\overline{w^{-1}}} - 1\right) + 1 \qquad (3)$$

where $\overline{w^{-1}}$ is the spatial average of the inverse actual speed $w(x,y)$. Plotting $\rho^*$ as a function of $1/V$ we find that this simple model quantitatively explains the density behavior in **Figure 3** with the natural choice of free parameters $\alpha = 0.5$, $\beta = 0.44$, and $k^{-1} = \overline{v}\tau_m = 59$ μm where $\overline{v}$ is the spatial average of $V(x,y)$. This result is certainly encouraging and prompts for a deeper and more systematic investigation using simpler light patterns. Our aim here, however, is to investigate the resolution limits of density shaping with photokinetic bacteria. Although memory effects could be reduced by deleting ATP-synthase genes (**Arlt et al., 2018**), resulting in a suppression of the slow component in speed relaxation, response will never be truly instantaneous. In addition to memory effects, steric and/or hydrodynamic interactions will always affect density by hindering bacteria penetration and further accumulation in densely packed regions. The combined action of these effects on the stationary density map cannot be easily incorporated in a theoretical model that provides precise quantitative predictions with a priori known parameters.

An alternative and more practical strategy to improve the accuracy of density shaping, is to implement a feedback control loop. We set up an automated control loop that performs one iteration every 20 s, comparing the current density map to the desired target image and updating the illumination pattern accordingly (see Materials and methods). Light levels are increased in those regions with a density that exceeds the target value and decreased where density is lower. At each step the light intensity increment is simply proportional to the pixel by pixel difference of target and actual density images. We quantify the distance between the obtained density pattern and the target image by the sum of the squared pixel-by-pixel difference after a proper rescaling procedure (see Materials and methods).

As soon as the feedback is turned on at time $t_0 = 6$ min (**Figure 5(a)**) the distance from the target starts to decrease reaching a new stationary value after about 4 min. The final stationary density map shows a level of detail significantly higher than the image obtained before the feedback was turned on (**Figure 5(b,c)**). The image is stable and resolution does not deteriorate for hours. However a dark region surrounds the illuminated area so that swimming cells that exit the field of view do not come back and the number of motile cells in the illuminated area is reduced by about 50% in one hour. This issue is the main limiting factor for the lifetime of our patterns and could be solved in the future by using sample chambers that match the size of the illuminated area. As evidenced in **Figure 5(d)**, the feedback precompensates this blurring effect due to memory by converging to a sharpened version of the initial target image. Results of comparable quality have been replicated four times in each of the two independent biological replicates.

A remarkable feature of this optically controlled material lays in its intrinsic dynamic and reconfigurable nature. As a demonstration of this property we show the dynamic morphing of a bacterial layer from an Albert Einstein's to a Charles Darwin's portrait (see **Figure 6** and **Video 1**). The morphing is triggered by instantly switching between the two target images on the DLP projector and exponentially converges to the second target in a characteristic time of 1.8 min.

## Discussion

Although light induced density modulations can be achieved in large samples of Brownian (passive) colloidal systems, active materials provide a largely superior performance in terms of both required power and response time. The stationary density distribution of Brownian colloids is governed by the Boltzmann law stating that the logarithmic ratio of maximum to minimum density is given by

$$\log\left(\frac{\rho_{\max}}{\rho_{\min}}\right) = -\frac{\Delta U}{k_B T},$$

where $\Delta U$ is the optical energy difference between brightest and darkest regions. For a plastic or glass bead of radius $a = 1\,\mu m$ in water, this energy difference can be estimated in the Rayleigh regime as (*Ashkin et al., 1986*) $\Delta U \approx a^3 I/c$ where $I$ is the maximum power density and $c$ is the speed of light. For a contrast level comparable to the ones shown above, $\log(\rho_{\max}/\rho_{\min}) \approx 1$, the required power density would be $I \approx k_B T c/a^3 \approx 1$ W/mm$^2$. This value is three orders of magnitude larger than the maximum power densities used in our experiments. Our active system is also much faster than its Brownian counterprt as evidenced by comparing the timescales of pattern formation. In both the active and passive case, dynamics is governed by the interplay of diffusion and drift with characteristic timescales $\tau_{\mathrm{diff}}$ and $\tau_{\mathrm{drift}}$. Calling $\ell = 1$ mm the largest length scale in the target pattern, we obtain for the active case $\tau_{\mathrm{drift}} \approx \ell/v \approx 200$ s where we used $v = 5\,\mu m/s$ as the typical swimming speed. A diffusion time scale can be obtained as $\tau_{\mathrm{diff}} \approx \ell^2/2D$ where $D$ is the active translational diffusion coefficient. For wild type bacteria, moving with a run and tumble dynamics, $D = v^2 \tau_{run}$ where $\tau_{run} \approx 1$ s is the mean duration of a run. The corresponding characteristic diffusion time is $\tau_{\mathrm{diff}} \approx \ell^2/2D \approx 2 \times 10^4$ s. This timescale can be considerably reduced if, as in our case, we suppress tumbling and use a smooth swimming strain with a much larger reorientation time $\tau_{\mathrm{rot}} \approx 20$ s (*Berg, 1993*). In this case the diffusion timescale drops down to $\tau_{\mathrm{diff}} \approx 10^3$ s leading to a faster relaxation towards the stationary state. In the Brownian case the drift timescale is given by $\tau_{\mathrm{drift}} \approx \mu k_B T/\ell \approx 10^6$ s where $\mu = 50\,\mu m/s$ pN is the mobility of a 1 µm radius microsphere. The same colloidal particle has a diffusivity $D = \mu k_B T = 0.2$ µm$^2$/s giving again $\tau_{\mathrm{diff}} \approx \ell^2/2D \approx 10^6$ s. Summarizing, using optical forces to achieve a comparable density modulation for Brownian particles would require a thousand times larger powers and from $10^3$ to $10^4$ longer times.

In conclusion, we have shown that a suspension of swimming bacteria, with optically controllable speed, can provide a new class of light controllable active materials whose density can be accurately, reversibly and quickly shaped by employing a low power light projector. An alternative strategy to produce static patterns of bacteria is by 'biofilm lithography' where an optical template of blue light is used to induce the expression of membrane proteins that promote cell-cell and cell-substrate attachment (*Jin and Riedel-Kruse, 2018*). As opposed to our motility driven density shaping, these patterns are static and form over a time scale of several hours. Interestingly, the two techniques could be used in combination to achieve faster and more complex (e.g. three dimensional) biofilm lithography by fixing with blue light a template pattern obtained by speed modulation with green light. By further genetic engineering, bacteria could be eventually encapsulated in silicate shells (*Müller et al., 2008*) producing solid permanent structures for micro-mechanics or micro-optics applications. Finally, the possibility of spatial and temporal control of motile bacteria density could also lead to novel strategies for the transport and manipulation of small cargoes inside microdevices *Maggi et al. (2018)*.

## Materials and methods

### Production of photokinetic cells

For all the experiments we used the *E. coli* strain HCB437 (*Wolfe et al., 1987*) transformed with a plasmid encoding the PR under the control of the araC-pBAD promoter (Biobricks, BBa_K1604010 inserted in pSB1C3 plasmid backbone). *E. coli* colonies from frozen stocks are grown overnight at 33°C on LB agar plates supplemented with kanamycin (Kan 30 µg/mL) and chloramphenicol (Cam 20 µg/mL). A single colony is picked and statically cultivated overnight at 33°C in 10 mL of M9 broth (M9 salts with 0.2% glucose, 0.2% casaminoacids) supplemented with antibiotics as before. The overnight culture is diluted 100-fold into 5 mL of the previous medium, grown at 33°C, 200 rpm. 5 mM

arabinose and 20 µM retinal are added once $OD_{590} \approx 0.2$, keeping the culture in the dark to avoid retinal degradation. Once $OD_{590} \approx 0.8$, cells are collected by centrifugation (1500 rcf, 5'). The resulting pellet is washed twice by centrifugation (1500 rcf, 5') with motility buffer (MB: 0.1 mM EDTA, 10 mM phosphate buffer and 0.2% Tween20). The bacterial suspension is eventually adjusted to a working $OD_{590} \approx 2.0$.

## Sample preparation

100 µL of the prepared bacterial suspension is injected into a glass capillary (CM Scientific - Rect. boro capillaries 0.40 × 4 mm), sealed at both sides with vacuum grease (Sigma-Aldrich). The capillary is then placed under the optical microscope and observed in bright and dark field illumination focusing on the plane in the middle of the capillary.

## Optics

Bright field and dark field imaging were performed using a custom inverted optical microscope equipped with a 4 × magnification objective (Nikon; NA = 0.13) and a high-sensitivity CMOS camera (Hamamatsu Orca-Flash 2.8) (see *Figure 1*). Light shaping was performed using a digital light processing (DLP) projector (Texas Instruments DLP Lightcrafter 4500) coupled to the same microscope objective used for imaging. The size of a (squared) DLP projector pixel imaged on the sample plane results to be 2 µm. All dark-field images have been divided by the flat field image obtained by applying a Gaussian filter on the respective homogeneous bacterial suspension (under uniform green illumination) as detailed in the following. Flat back-ground subtraction (fifth percentile of the image histogram) was only applied to the frames appearing in *Figure 6*.

## Differential Dynamic Microscopy (DDM)

To perform DDM we compute the quantity:

$$g(\mathbf{q}, t', t) = \langle |M(\mathbf{q}, t') - M(\mathbf{q}, t' + t)|^2 \rangle \qquad (4)$$

where $M(\mathbf{q}, t)$ is the spatial Fourier transform at the wave-vector $\mathbf{q}$ of the bright-field image captured at time $t$. Assuming time-translational invariance and isotropy in the bacterial movement $g(\mathbf{q}, t', t)$ depends only on the time-lag $t$ in *Equation (4)* and on the modulus of the wave vector $q = |\mathbf{q}|$, that is $g(\mathbf{q}, t, t') = g(q, t)$. The $g(q, t)$ is connected to the intermediate scattering function (ISF) $F(q, t)$:

$$g(q, t) = A(q)F(q, t) + B(q) \qquad (5)$$

where $A(q)$ and $B(q)$ are time-independent factors related, respectively, to the number and shape of bacteria, and to the background noise in the images. Following *Wilson et al., 2011* we use the ISF model for independent smooth swimming cells:

$$F(q, t) = (1 - \alpha)e^{-q^2 Dt} + \alpha e^{-q^2 Dt} \int_0^\infty dv' \, P(v') \, \mathrm{sinc}(qv't) \qquad (6)$$

where $\alpha$ is the fraction of motile cells, $D$ the Brownian diffusion coefficient and $P(v')$ the Schultz distribution:

$$P(v') = \frac{\frac{1}{v'} \left(\frac{Z+1}{v} v'\right)^{Z+1} \exp\left(-\frac{Z+1}{v} v'\right)}{\Gamma(Z+1)} \qquad (7)$$

Here $v$ is the mean speed, $\Gamma$ is the Euler gamma function and $Z$ is related to the speed standard deviation $\sigma$ by the formula $Z = (v/\sigma)^2 - 1$. The relevant parameters discussed in the main text $v$ and $\sigma$ are extracted by fitting the experimental $g(q, t)$ in the $q$-range 0.45 µm$^{-1}$<q<1.2 µm$^{-1}$ with *Equations 5, 6 and 7*.

## Image analysis

We assume that dark field images are proportional to the bacterial density modulated by a slowly varying envelope due to inhomogeneities in illumination. This flat field correction $\rho_0(\mathbf{r})$ is obtained by acquiring 100 frames at 25 fps at the beginning of the experiments when the bacterial density is

homogeneous. These frames are averaged and then filtered with a Gaussian kernel with a large standard deviation ($\approx 100\,\mu\text{m}$). The normalized experimental density $\rho^*(\mathbf{r})$ is obtained by first computing a raw density field $\rho(\mathbf{r})$ (50 frames average at 25 fps) and then dividing by $\rho_0$, that is $\rho^*(\mathbf{r}) = \rho(\mathbf{r})/\rho_0(\mathbf{r})$.

For computing the distance between $\rho^*(\mathbf{r})$ and the target density $\rho^{\text{tar}}(\mathbf{r})$ we first rescale $\rho^*(\mathbf{r})$ so that the 10-th and 90-th percentile of the two image histograms coincide:

$$\rho^{\text{s}}(\mathbf{r}) = \frac{\rho^{\text{tar}}_{90\%} - \rho^{\text{tar}}_{10\%}}{\rho^*_{90\%} - \rho^*_{10\%}}\left(\rho^*(\mathbf{r}) - \rho^*_{10\%}\right) + \rho^{\text{tar}}_{10\%}.$$

The distance *dist* is then computed as:

$$dist = \left[\sum_{\mathbf{r}}\left[\rho^{\text{s}}(\mathbf{r}) - \rho^{\text{tar}}(\mathbf{r})\right]^2\right]^{\frac{1}{2}}$$

where the sum runs over the image pixels $\mathbf{r}$.

## Modeling the effects of memory in speed response

A smooth swimming cell with a two-step response to light intensity and traveling in the $x$ direction over a light imposed speed profile $V(x)$, will have an actual speed at time $t$ given by:

$$v(t) = \beta V(x(t)) + (1-\beta)\int_0^{\infty} V(x(t-t'))\,e^{-t'/\tau_m}dt' \tag{8}$$

where $x(t)$ is the position of the cell at time $t$. For a weakly varying speed profile $V(x) = V_0 + \delta V(x)$ we can transform the above time convolution into a space convolution by assuming $x(t) \approx x(0) + V_0 t$:

$$v(x) \simeq \beta V(x) + (1-\beta)\frac{k}{2}\int_{-\infty}^{\infty} V(x-s)e^{-k|s|}ds \tag{9}$$

where we have also assumed that an equal number of bacteria will be traveling in the opposite direction. Assuming isotropy and using the fact that, in the weak modulation limit, density is homogeneous at the zero order, we can easily generalize to the three dimensional case and write:

$$v(\mathbf{r}) \simeq \beta V(\mathbf{r}) + (1-\beta)\frac{k}{4\pi}\int V(\mathbf{r}-s\hat{u})e^{-k|s|}dsd\Omega = \beta V(\mathbf{r}) + (1-\beta)\int V(\mathbf{r}-\mathbf{r}')\Gamma(\mathbf{r}')d^3r' \tag{10}$$

where $\Gamma(\mathbf{r}) = (k/4\pi r^2)\,e^{-kr}$ is a 3D convolution kernel. Our projecting light system is such that light patterns do not vary significantly across the entire sample depth so that we can assume $V(x,y,z) = V(x,y)$ and express the effective speed distribution as a 2D convolution

$$v(x,y) \simeq \beta V(x,y) + (1-\beta)\int V(x-x',y-y')\gamma(x',y')dx'dy' \tag{11}$$

with $\gamma(x,y) = \int_{-\infty}^{\infty}\Gamma(x,y,z)dz$. Although the convolution kernel $\gamma$ is not analytic in real space, its Fourier transform is analytic making convolutions easy to calculate numerically:

$$\widetilde{\gamma}(q) = \frac{k}{q}\arctan\left(\frac{q}{k}\right) \tag{12}$$

where $q = \sqrt{q_x^2 + q_y^2}$.

## Feedback

The $(n+1)$-th illumination pattern at the pixel $\mathbf{r}$ is updated as follows:

$$I_{n+1}(\mathbf{r}) = I_n(\mathbf{r}) + P\Delta\rho(\mathbf{r}),$$

where $P>0$ is a proportional control parameter and $\Delta\rho$ is the difference between the scaled density and the target density:

$$\Delta\rho(\mathbf{r}) = \rho^{\mathrm{s}}(\mathbf{r}) - \rho^{\mathrm{tar}}(\mathbf{r})$$

If, for example, the scaled density is larger than the target at $\mathbf{r}$ the projected power density at that pixel will be increased. That will increase the speed of bacteria and consequently reduce their local density.

## Acknowledgements

The research leading to these results has received funding from the European Research Council under the European Union's Seventh Framework Programme (FP7/2007-2013)/ERC grant agreement no. 307940.

## Additional information

### Funding

| Funder | Grant reference number | Author |
| --- | --- | --- |
| European Research Council | 307940 | Roberto Di Leonardo |

The funders had no role in study design, data collection and interpretation, or the decision to submit the work for publication.

### Author contributions

Giacomo Frangipane, Conceptualization, Software, Formal analysis, Investigation, Methodology, Writing—original draft; Dario Dell'Arciprete, Investigation, Methodology, Writing—original draft; Serena Petracchini, Investigation, Methodology; Claudio Maggi, Conceptualization, Formal analysis, Investigation, Writing—original draft; Filippo Saglimbeni, Data curation, Methodology; Silvio Bianchi, Gaszton Vizsnyiczai, Software, Methodology; Maria Lina Bernardini, Supervision, Methodology; Roberto Di Leonardo, Conceptualization, Formal analysis, Supervision, Investigation, Methodology, Writing—original draft

### Author ORCIDs

Giacomo Frangipane http://orcid.org/0000-0002-1533-4754
Claudio Maggi https://orcid.org/0000-0003-0033-6287
Roberto Di Leonardo http://orcid.org/0000-0002-5020-0663

### Decision letter and Author response

Decision letter https://doi.org/10.7554/eLife.36608.017
Author response https://doi.org/10.7554/eLife.36608.018

## Additional files

### Supplementary files

• Transparent reporting form
DOI: https://doi.org/10.7554/eLife.36608.015

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
