## [Decision Letter]

Thank you for submitting your article "Dynamic density shaping of photokinetic *E. coli*" for consideration by *eLife*. Your article has been reviewed by three peer reviewers, including Raymond E Goldstein as the Reviewing Editor and Reviewer #1, and the evaluation has been overseen by Naama Barkai as the Senior Editor.

The reviewers have discussed the reviews with one another and the Reviewing Editor has drafted this decision to help you prepare a revised submission.

Summary:

In this paper, the authors demonstrate the ability to dynamically arrange large collections of *E. coli* bacteria into complex two-dimensional density patterns using light illumination using a digital projector. The method relies on *E. coli* mutants whose speed depends on the local level of illumination and is based on a previously proposed relationship between swimming speed and density in suspensions of self-propelled particles. The authors further implement a feedback control strategy to improve resolution, where they adjust illumination based on current density. Their method is shown to be rapid (compared to methods based on passive colloids), to be able to capture fine spatial features, and to be reversible.

Essential revisions:

1) The role of run-and-tumble locomotion in this whole process should beexplained. Perhaps the authors could clarify the length/time scales associated with translational diffusion coming from that process in the directed pattern formation.

2) A more detailed discussion of resolution (and what controls/limits resolution) would be useful.

3) Presentation: Many concepts and methods warrant expanded explanation. Symbols are introduced without definition. There is inadequate description of experimental methods, a diagram of the experimental protocol would have been helpful, highlighting parameters such as size and geometry of chamber etc. The method of preparation of the bacterial suspension is also unclear. (In fact, supplementary figure 1 might be helpful as part of the main text.) Several algorithms for data extraction or image manipulation are glossed over and require careful definition.

4a) Interpretation: Discussion of the theoretical background can be improved. For instance, the assumed inverse proportionality law between local density and speed is itself a simplification, and the limitations of this simple theory could be discussed in the context of the experimental results. (The discussion of Figure 1B is ambiguous.)

4b) There is a growing body of work in which active matter is manipulated successfully with light, and these should be cited and discussed. Notably, recent papers from the Riedel-Kruse lab (Jin and Riedel-Kruse, 2018, PNAS), or the Poon group (Arlt et al., 2018, Nature Comms.), and several others besides. The authors should ensure previously published literature is clearly referenced, and importantly, how their work differs from or represents improvements upon these previous results.

5) Purpose: While this is a beautiful result, the wider significance of this work is unclear. We are unconvinced by the bioengineering argument that these active materials are somehow better/more controllable for micro optics applications; it is argued in terms of drift and diffusion timescales that active systems are much more responsive than their Brownian counterparts, this is undeniable – but why might it be desirable to control such samples in the first place? Furthermore, are these patterns stable? If continued to be subjected to the light, individual bacteria will adapt to the surrounding light conditions, how long will a typical Mona Lisa image remain recognisable before the pattern degrades?

We would have liked to have seen either (i) a clear theoretical interpretation in terms of the theory of motility induced phase separation, e.g. what are the physical reasons for the measured deviations from the simple inverse density-speed relationship, or (ii) a clear biological interpretation of the timescales/transients involved in the pattern formation process in terms of some intracellular "memory". More interesting would be if the authors could measure, by combining precisely projected patterns with feedback modulation of population density, the actual spatiotemporal memory of the bacteria and how this might change with genetic manipulation.

---

## [Author Response]

Essential revisions:1) The role of run-and-tumble locomotion in this whole process should beexplained. Perhaps the authors could clarify the length/time scales associated with translational diffusion coming from that process in the directed pattern formation.

We thank the reviewers for pointing that out. We now discuss translational

diffusion timescales for wild type (run and tumble) bacteria which

gives us the opportunity of highlighting the importance of using smooth

swimmers for faster density shaping. This is discussed in the revised

manuscript as follows:

“Calling *ℓ* = 1 mm the largest length scale in the target pattern, we obtain for the active case *τ*_drift_ ≈ *ℓ/v* ≈ 200 s where we used *v* = 5 μm/s as the typical swimming speed. A diffusion time scale can be obtained as *τ*_diff_ ≈ *ℓ*^2^*/*2*D* where *D* is the active translational diffusion coefficient. For wild type bacteria, moving with a run and tumble dynamics, *D* = *v*^2^*τ*_run_ where *τ*_run_ ≈ 1 s is the mean duration of a run. The corresponding characteristic diffusion time is *τ*_diff_ ≈ *ℓ*^2^*/*2*D* ≈ 2 × 10^4^ s. This timescale can be considerably reduced if, as in our case, we suppress tumbling and use a smooth swimming strain with a much larger reorientation time *τ*_rot_ ≈ 20 s (Ashkin et al., 1986). In this case the diffusion timescale drops down to *τ*_diff_ ≈ 10^3^ s leading to a faster relaxation towards the stationary state.”

2) A more detailed discussion of resolution (and what controls/limits resolution) would be useful.

We now discuss resolution limits in greater detail:

“To validate this hypothesis we use a DLP projector to display a complex light pattern onto a 400 μm thick layer of cells (Figure 1) that have been preliminarily exposed to a uniform bright illumination for 5 min. This time is much longer than the speed response time of bacteria (Figure 4) thus ensuring that cells are initialized to swim at maximal speed. The projecting system has an optical resolution of 2 μm approximately matching the size of a single cell which represents the physical “pixel” of our density images. This value sets the limit for the minimum theoretical resolution of density configurations that would be achievable if bacteria could be precisely and statically arranged in space. In practice, as we will see, the real resolution will always be larger for two main reasons: (i) bacteria do not respond instantly to light temporal variations thus introducing a blur in the target speed map, (ii) the stationary state is an ensemble of noisy patterns that constantly fluctuate because of swimming and Brownian motions of bacteria.”

See also reply to point 4 for a deeper theoretical analysis of how memory affects resolution and how feedback can help alleviating this blurring effect.

3) Presentation: Many concepts and methods warrant expanded explanation. Symbols are introduced without definition. There is inadequate description of experimental methods, a diagram of the experimental protocol would have been helpful, highlighting parameters such as size and geometry of chamber etc. The method of preparation of the bacterial suspension is also unclear. (In fact, supplementary figure 1 might be helpful as part of the main text.) Several algorithms for data extraction or image manipulation are glossed over and require careful definition.

We thank the reviewers for addressing these issues. We fully agree with the above comments and have extensively expanded the Materials and methods section to describe the details of data extraction and image analysis.

The Materials and methods section now includes a detailed subsection about DDM and analysis.

We have also added a detailed explanation of the normalization procedure, the computation of the distances from the target and the feedback loop (subsection “Image analysis” and subsection “Feedback”).

We also moved the supplementary figure to the main text as Figure 1 after adding schematic and geometric details of sample preparation.

4a) Interpretation: Discussion of the theoretical background can be improved. For instance, the assumed inverse proportionality law between local density and speed is itself a simplification, and the limitations of this simple theory could be discussed in the context of the experimental results. (The discussion of Figure 1B is ambiguous.)

We thank the reviewers for pushing us to go further in this direction. Stimulated by their questions we arrived at an original theoretical result that provides an expression for the stationary density of smooth swimming bacteria exploring a non-homogeneous light pattern and with a speed response to temporal light changes that is not instantaneous. The details of the theory are reported in the Materials and methods section. The revised manuscript now includes a discussion of this theoretical result in the context of experimental data:

“As a result, bacteria will experience an effective speed map that is a blurred version of what we would expect for an instantaneous response *V* (**r**) = *v(I*(**r**)). […]

Our aim here, however, is to investigate the resolution limits of density shaping with photokinetic bacteria.”

4b) There is a growing body of work in which active matter is manipulated successfully with light, and these should be cited and discussed. Notably, recent papers from the Riedel-Kruse (Jin and Riedel-Kruse, 2018, PNAS), or the Poon group (Arlt et al., 2018, Nature Comms.), and several others besides. The authors should ensure previously published literature is clearly referenced, and importantly, how their work differs from or represents improvements upon these previous results.

The paper by the Poon group was already referenced (Arlt et al., 2018). We now better highlight the main differences from our work as follows:

“[1] have recently shown that, by projecting a masked illumination pattern, “photokinetic” [11] bacteria can be accumulated in dark regions and depleted from brighter ones thus forming binary patterns. […] Finally we show that a model independent feedback control loop allows density shaping with high spatial resolution and gray level accuracy.”

We thank the reviewers for pointing us to the paper by the Reidel-Kruse lab which we have included in the bibliography and discussed in the concluding section:

“An alternative strategy to produce static patterns of bacteria is by “biofilm lithography” where an optical template of blue light is used to induce the expression of membrane proteins that promote cell-cell and cell-substrate attachment [4]. As opposed to our motility driven density shaping, these patterns are static and form over a time scale of several hours. Interestingly, the two techniques could be used in combination to achieve faster and more complex (e.g. three dimensional) biofilm lithography by fixing with blue light a template pattern obtained by speed modulation with green light.”

5) Purpose: While this is a beautiful result, the wider significance of this work is unclear. We are unconvinced by the bioengineering argument that these active materials are somehow better/more controllable for micro optics applications; it is argued in terms of drift and diffusion timescales that active systems are much more responsive than their Brownian counterparts, this is undeniable – but why might it be desirable to control such samples in the first place? Furthermore, are these patterns stable? If continued to be subjected to the light, individual bacteria will adapt to the surrounding light conditions, how long will a typical Mona Lisa image remain recognisable before the pattern degrades?

We have expanded the concluding section including novel application ideas:

“In conclusion, we have shown that a suspension of swimming bacteria, with optically controllable speed, can provide a new class of light controllable active materials whose density can be accurately, reversibly and quickly shaped by employing a low power light projector. […] Finally, the possibility of spatial and temporal control of motile bacteria density could also lead to novel strategies for the transport and manipulation of small cargoes inside microdevices [5].”

Regarding stability, as we now say in the main text:

“The image is stable and resolution does not deteriorate for hours. However a dark region surrounds the illuminated area so that swimming cells that exit the field of view do not come back and the number of motile cells in the illuminated area is reduced by about 50% in one hour. This issue is the main limiting factor for the lifetime of our patterns and could be solved in the future by using sample chambers that match the size of the illuminated area.”

We would have liked to have seen either (i) a clear theoretical interpretation in terms of the theory of motility induced phase separation, e.g. what are the physical reasons for the measured deviations from the simple inverse density-speed relationship, or (ii) a clear biological interpretation of the timescales/transients involved in the pattern formation process in terms of some intracellular "memory". More interesting would be if the authors could measure, by combining precisely projected patterns with feedback modulation of population density, the actual spatiotemporal memory of the bacteria and how this might change with genetic manipulation.

We decided to move in the second direction and investigate the role of intracellular “memory” with new experiments and an original theoretical analysis. We have used DDM to accurately evaluate the time response of swimming speed to a square wave light intensity modulation. Obtained data are now reported in a new figure and discussed in the main text as follows:

“In particular, the theory assumes that speed is a local function of space which would only be the case if bacteria instantly adapt to temporal changes in light intensity. […] A slower relaxation follows and it is well fitted by an exponential function with a time constant *τ*_m_ = 35 s that is the same for both the rising and falling relaxations.”

Looking for a possible origin of the slow component we have performed a more detailed genetic screening of our strain. In particular, contrary to our original expectations, we confirmed with PCR the presence of the *unc*-operon coding for the ATP-synthase complex. The Materials and methods section has been modified accordingly.